# Synthesis and Photophysical Properties of α-(*N*-Biphenyl)-Substituted 2,2′-Bipyridine-Based Push–Pull Fluorophores

**DOI:** 10.3390/molecules27206879

**Published:** 2022-10-13

**Authors:** Ekaterina S. Starnovskaya, Dmitry S. Kopchuk, Albert F. Khasanov, Olga S. Taniya, Igor L. Nikonov, Maria I. Valieva, Dmitry E. Pavlyuk, Alexander S. Novikov, Grigory V. Zyryanov, Oleg N. Chupakhin

**Affiliations:** 1Chemical Engineering Institute, Ural Federal University, 19 Mira St., Yekaterinburg 620002, Russia; 2I. Ya. Postovsky Institute of Organic Synthesis of RAS, Ural Division, 22/20 S. Kovalevskoy/Akademicheskaya St., Yekaterinburg 62099, Russia; 3Institute of Chemistry, Saint Petersburg State University, 7/9 Universitetskaya Nab., Saint Petersburg 199034, Russia; 4Research Institute of Chemistry, Peoples’ Friendship University of Russia (RUDN University), 6 Miklukho-Maklaya St., Moscow 117198, Russia

**Keywords:** α-(*N*-Biphenyl)-substituted 2,2′-bipyridines, “push–pull” fluorophores, fluorescence, fluorosolvatochromic properties, Lippert–Mataga equation, AIEE, Hammett constants, nitroexplosive detection

## Abstract

A series of new α-(*N*-biphenyl)-substituted 2,2′-bipyridines were obtained through the combination of the ipso-nucleophilic aromatic substitution of the C5-cyano group, aza-Diels–Alder and Suzuki cross-coupling reactions, starting from 5-cyano-1,2,4-triazines. For the obtained compounds, photophysical and fluorosolvatochromic properties were studied. Fluorophores **3l** and **3b** demonstrated unexpected AIEE activity, while **3a** and **3h** showed promising nitroexplosive detection abilities.

## 1. Introduction

Compounds based on α-arylamino-2,2′-bipyridines and their fused analogs have already found wide practical applications. In particular, they serve as promising drug candidates, such as tyrosine kinases [1,2,3] or cyclooxygenase [4] modulators, and as promising candidates for the treatment of Lysosomal Storage Diseases (LSDs) [5] and other applications. In addition, α-arylamino-2,2′-bipyridines and their derivatives were reported as ligands for the transition metal complexes exhibiting promising photoluminescent behavior [6,7,8,9]. Several *N*-arylamino-substituted 2,2′-bipyridines reported as components for electroluminescent devices [10] and OLED systems [11]. In addition, platinum (II) or palladium (II) complexes based on *N*-aryl-(2,2′-bipyridin-6-yl)amines were reported as reagents for the photodynamic therapy of lung and prostate cancers [12,13] (Figure 1). Several isomeric amino-substituted 2,2′-bipyridines were reported as fluorescent indicators for Zn (II) and fluorescent dyes for the imaging of prostate cells [14]. Finally, arylamino-2,2′-bipyridines were reported as convenient synthons for the construction of various fused polynuclear systems [15,16,17].

Despite the extensive scope of practical applications for arylamino-substituted 2,2′-bipyridines, approaches for their synthesis are very limited. For instance, amination reactions via ipso-nucleophilic aromatic substitution of leaving groups, which are mainly halogen atoms, have become a most widely used methodology. Thus, interactions of mono- and dichloro-1,10-phenanthrolines with aniline or p-phenylenediamine have been described [16,18]. Another approach involves a deoxygenative amination of azine-*N*-oxides with acyl azides via a [3 + 2] cycloaddition reaction [19]. Finally, the Pd-catalyzed Buchwald–Hartwig amination reaction was involved in case of interactions between bromine-/chloro-substituted bipyridines or terpyridines and aromatic (di)amines. As a result, various mono- and bis-arylamino-2,2′-bipyridines were obtained [6,13,17,20].

Recently, our group reported a more rational methodology for the preparation of α-arylamino-2,2′-bipyridines as promising push–pull fluorophores [21,22,23]. This methodology involved a transition metal-free (TM-free) direct ipso-amination of 3-(pyridin-2-yl)-5-cyano-1,2,4-triazines under the action of various anilines, including fluorine-substituted ones, in solvent-free conditions upon heating. A subsequent solvent-free aza-Diels–Alder reaction between C5-arylamino-substituted 1,2,4-triazines with 1-morpholinecyclopentene and 2,5-norbornadiene leads to the formation of target arylamino-substituted 2,2′-bipyridine. In this work, we wish to report a modification of the above-mentioned approaches to introduce the residues of [1,1′-biphenyl]-4-amine into the 1,2,4-triazine core [24] with the following, obtaining α-(*N*-biphenyl)-substituted 2,2′-bipyridines as push–pull fluorophores.

## 2. Results and Discussion

### 2.1. Synthesis of α-(N-Biphenyl)-Substituted 2,2′-Bipyridines ***3a**–**l***

As a first step, *N*-(biphenyl-3-yl)-1,2,4-triazines **1a**–**c** were prepared in 84–88% yields according to a previously described procedure [24] through ipso-nucleophilic aromatic substitution of the C5-cyano group in the reaction of 5-cyano-1,2,4-triazines **2a**,**b** and [1,1′-biphenyl]-3-amines in solvent-free conditions. The following aza-Diels–Alder reaction between them thus obtained C5-amino-substituted 1,2,4-triazines **1** with 1-morpholinocyclopentene-afforded cyclopentane-fused 2,2′-bipyridines **3a**–**c** in 67–74% yields (Figure 1).

Some limitations of the herein-reported method should be mentioned. Previously we were unable to introduce residues of [1,1′-biphenyl]-4-amines **4a**,**b** into the 1,2,4-triazine core [24]. A possible reason for the low reactivity of 4-aminobiphenyls is the high delocalization of the electron pair of the nitrogen atom due to the resonance with the extended polyaromatic system. In this regard, we developed an alternative approach towards α-arylamino-2,2′-bipyridines **3a**–**l** based on the use of commercially available 3- and 4-bromoanilines. As a result, the ipso-nucleophilic aromatic substitution of the C5-cyano group in 1,2,4-triazines resulted in the formation of intermediate products **5a**–**c**, while subsequent aza-Diels–Alder reactions resulted in bromoarene-substituted 2,2′-bipyridines **6a**–**c**. The final products **3a**–**l** were obtained due to Suzuki cross-coupling reactions between 2,2′-bipyridines **6a**–**c** and the corresponding arylboronic acid. In this case, the choice of the solvent system turned out to be important for the success of the cross-coupling reaction. In particular, it was not possible to satisfactorily obtain product **3** by a fairly traditional mixture of toluene, water, and ethanol, and a large number of unidentified impurities were presented in the reaction mixture. However, in a THF:water = 1:1 mixture this reaction proceeded smoothly to afford the desired products in up to 74% yields.

The structures of the products **3a**–**l** were confirmed by the data of ^1^H and ^13^C NMR spectroscopy, as well as mass spectrometry and elemental analysis. Thus, in ^1^H NMR spectra of the obtained products, the signals of protons of the 2-pyridyl residue, bipyridine aromatic substituents and biphenyl fragments, as well as a broadened singlet of the N-H group proton in the region of 6.72–7.04 ppm, were observed. In addition, the structure of key 2,2′-bipyridine **3a** was confirmed by means of single crystal X-ray diffraction analysis (Figure 2). According to the X-ray data, compound **3a** crystallizes in the centrosymmetric space group of the monoclinic system. The bond distances and angles in the molecule are near standard values. The molecule is nonplanar and only the bipyridine atoms are placed in a plane with trans-placed nitrogen atoms. The nitrogen atom of the diarylamino-moiety is planar and the NH-group is not involved in the formation of the H-bond. No significantly shortened intermolecular contacts in the crystal were observed.

### 2.2. Photophysical Studies

As a next step, the photophysical properties of the obtained α-biphenylamino-2,2′-bipyridines **3a**–**l** were studied (Table 1 and Appendix A; Figure 3 and Appendix A). These compounds exhibited an intense blue-to-green fluorescence with emission maxima in the range of 443–505 nm and fluorescence quantum yields up to 49.1% in THF solutions. Compared to previously described unsubstituted analogs (Table 1, Entries 1–4 and 12–15), as well as 3- and 4-bromophenyl-substituted precursors (Entries 11 and 22, 23), the extension of the conjugation system in the aniline moieties in **3a**–**l** expectedly redshifted the emission maxima with the highest values in the case of 4-biphenylamino derivatives. The Stokes shift values are either the same level as described earlier, or significantly exceed them and can reach up to 9670.29 cm^−1^ or 169 nm (Entry 10). This difference is very significant compared to α-phenylamino 2,2′-bipyridines (Entry 1 and 12). The highest quantum yields among all the products were observed for compound **3a** (49%, Entry 5), **3b** (48%, Entry 16), and **3d** (44%, Entry 18), functionalized at the *meta*-position of the aniline unit. These values are comparable with the quantum yields of α-arylamino-2,2′-bipyridines that are not functionalized with additional aromatic moieties. However, a significant decrease in quantum yields was observed in the case of some *para*-substituted derivatives (Entries 8–10).

It is worth noting the change in the photophysical properties of α-arylamino-2,2′-bipyridines bearing methoxy- (Entry 14) or 4-methoxyphenyl groups (Entry 21). In this case, the introduction of an additional 1,3-phenylene fragment into the bipyridine core results in a blueshift of the emission maximum by 19 nm and the reduction of the Stokes shift value (7543.99 cm^−1^ vs. 7056.13 cm^−1^). However, fluorescence quantum yield increased from 4% to 21%. Therefore, the introduction of additional aromatic substituents into the α-aniline fragment of the 2,2′-bipyridine core could be used as a tool to tune the photophysical properties and is of particular interest.

### 2.3. Studies of Fluorosolvatochromism

In our recent works, we have already demonstrated a well-pronounced fluorosolvatochromic behavior of 2,2′-bipyridines functionalized with donor substituents, including aniline residues [22,23,25,26]. The herein-reported compounds **3a–l** also exhibited a pronounced fluorosolvatochromism, which was demonstrated for the most representative compounds **3b**,**g**,**i**,**l** (Figure 4, Figure 5 and Appendix A).

Solvent polarity did not significantly affect the absorption spectra, while it influenced significantly the fluorescence spectra of the fluorophores **3b**,**g**,**i**,**l** (Figure 4, Figure 5 and Appendix A). Thus, a gradual increase of the solvent polarity from cyclohexane to DMSO led to a gradual redshift of the fluorescence maxima with an increase of the Stokes shift values. For example, the observed fluorescence maximum for **3i** in cyclohexane was 438 nm, and the Stokes shift was 99 nm, while in DMSO these values were 515 nm and 176 nm, respectively. In the case of compound **3l**, the use of more polar solvents (DMF, acetonitrile and methanol) resulted in a blueshift of the emission maxima relative to the emission in DMSO, while in a solvent row from cyclohexane to dichloromethane, an increase of the solvent polarity exhibited a redshift of the emission maxima. It is worth mentioning that the difference between the largest and smallest values of the emission maximum, in the case of **3b**,**g**,**i**,**l** (69 nm, 67 nm, 107 nm and 112 nm, respectively, Appendix A), is greater than the same difference (38–70 nm) for the previously reported aniline containing 2,2′-bipyridines [22,23].

To estimate the push–pull character of the obtained fluorophores, a mathematical analysis of the obtained experimental data was carried out using the Lippert–Mataga equation (Equation (1)) [27,28,29]. We found that the introduction of additional aromatic fragments into the aniline residue in compounds **3b**,**g**,**i**,**l** led to an increase in the difference between the dipole moments of the ground (*μ_G_*) and excited (*μ_E_*) states. As a result, we calculated Δμ for **3b**,**g**,**i**,**l** (Table 2). These values are relatively high for the 2,2′-bipyridine type fluorophores, second only to some examples (Appendix A) [22,25,26,30].
(1)$$v_{A}-v_{F}=\frac{2}{hc}\left(\frac{\varepsilon -1}{2\varepsilon +1}-\frac{n^{2}-1}{2n^{2}+1}\right)\frac{{\left(\mu_{E}-\mu_{G}\right)}^{2}}{a^{3}}$$

Equation (1). In the Lippert–Mataga equation (*ν_A_* and *ν_F_* are the wavenumbers (cm^−1^) of the absorption and emission, respectively, *h* is Planck’s constant, *c* is the speed of light in vacuum, *a* is the radius of the cavity in which the fluorophore resides (4Å, [27]), *μ_E_* and *μ_G_* are the excited and ground state dipole moment, respectively.

For a more detailed study of the effect of solvent polarity on the photophysical properties, a Lippert–Mataga plot was constructed, that is, the dependence of the Stokes shift (cm^−1^) of the fluorophore on the orientational polarizability of the solvent (Δf). This plot indicated that **3b**,**g**,**i**,**l** demonstrated a linear dependence (R^2^ > 0.90, Figure 6a and Appendix A) only in the range of cyclohexane—DCM solvents—while in more polar solvents, the linearity was broken and there was a sharp drop in the Stokes shift values (Figure 6a, Appendix A). Deviations from linearity in the Lippert–Mataga plots indicate the occurrence of some intermolecular interactions in these solvents, such as the hydrogen bonds formation, which is confirmed by a certain linearity in this case (R^2^ = 0.69 − 0.99, Appendix A), and can not be explained by traces of water in selected solvents due to the absence of these deviations in related α-arylamino-2,2**′**-bipyridines [22] (that is, the ones without the 1,3- or 1,4-phenylene unit). The protic nature of methanol could influence corresponding emission spectra as well. Thus, this unusual behavior is discussible.
(2)$$\Delta f=\frac{\varepsilon -1}{2\varepsilon +1}-\frac{n^{2}-1}{2n^{2}+1}$$

Equation (2). The orientational polarizability of the solvent, where *ε* and *n* are the dielectric constant and the refraction index for the solvents, respectively.

The fluorosolvatochromism of **3b**,**g**,**i**,**l** correlates well with the solvent polarity scales according to Kosower [31,32] and Dimroth/Reichardt [33,34] (Table 3, parameters Z and E_T_(30), respectively), while some deviations were observed only in the case of the most polar solvents (DMF—methanol). Nevertheless, Figure 6b indicates that the dependence of the fluorescence maxima on the values of the empirical polarity factor E_T_(30) proceeds linearly (R^2^ > 0.82, Appendix A), except for **3b**,**g**,**i** in methanol, as well as for **3l** in DMSO, DMF, acetonitrile, and methanol. This linear change in emission also confirms the phenomenon of fluorosolvatochromism (Table 3) [35].

The electron-donating or accepting ability of organic substituents is an important parameter affecting not only their reactivity but also the pKa of ionizable groups and chromophore properties. These substituent properties are usually described by Hammett sigma constants, constants obtained by the ionization of substituted benzoic acids. Although the values of these constants have been measured for the most common functional groups, data are not available for many important substituents. However, Peter Ertl published the results of a quantum chemical calculation of substituent descriptors compatible with Hammett sigma constants by using a more sophisticated methodology and a larger set of training data and presented a free web-based tool to calculate substituent descriptors compatible with Hammett sigma constants. This is available at: https://peter-ertl.com/molecular/substituents/sigmas.html (accessed on 16 May 2022) [36].

The dependence of emission maxima on Hammett sigma constants for two groups of α-biphenylamino-2,2′-bipyridine fluorophores, *para*- and *meta*-substituted with respect to the aniline residue, were plotted (Appendix A). Except for **3g**, *meta*-substituted fluorophores **3a**–**f** showed an acceptor character (Appendix A). At the same time, a good linear correlation with σ_m_ (R^2^ = 0.92) was observed provided that such strongest electron-withdrawing substituents as Cl (#3) and Br (**5c**) are included in the analyzed sequence. For donor-substituted compounds **3j**, **3k**, **#4**, **#14** including the hydrogen-substituted compound **3h**, as well as acceptor-substituted compounds **5a**, **5b**, **#15**, **#13**, a well-linear correlation with σ_p_ (R^2^ = 0.89) was observed (Appendix A). Thus, the correlation analysis of long-wavelength emission maxima plotted as a function of the Hammett constants made it possible to obtain the relationship between the structure and photophysical properties of the resulting fluorophores.

### 2.4. DFT Studies

To understand the structure-photophysical properties relationship, we performed density functional theory (DFT) calculations with the help of the Gaussian-09 [37] program package at the B3LYP/6-31G*//PM6 level in acetonitrile (see Computational Details and Appendix A) to obtain optimized molecular structures, energy levels of boundary molecular orbitals, dipole moments and molecular electrostatic potentials for the molecules **3b**,**g**,**i**,**l**. The HOMO-orbitals were distributed on the substituted biphenyl residues acting as electron donors, while LUMO-orbitals were distributed on the 2,2′-bipyridine cores acting as electron acceptors (Figure 7). This facilitates the intermolecular charge transfer (ICT) process (“push–pull” effect) in a molecule and correlates well with the photophysical experimental data. According to the calculation, the maximum difference between the dipole moments in the ground and excited states Δμ is more than 10D for the molecule **3l** (Appendix A), which is typical for the ICT state [38]. However, there are significant differences in the Δμ values between those calculated and obtained via Lippert–Mataga equations for the rest of molecules **3b**,**g**,**i** (Appendix A). The calculated values of the energy gaps (ΔE) correlate well with the UV spectroscopic data (Appendix A). Thus, a significant bathochromic shift of absorption maxima (289 nm and 342 nm) is observed in 4-diphenylaminophenyl-substituted compound **3l**, which is in a more favorable energy state (ΔE = 3.94 eV (Figure 7)).

In order to elucidate the electron transfer process for α-biphenylamino-2,2′-bipyridines, the molecular electrostatic potential (MEP) distribution (Appendix A) was calculated. The results obtained also confirm that the electron transfer occurs from an electron-rich biphenylene unit to the electron-deficient 2,2′-bipyridine core leading to the “push–pull” process.

### 2.5. AIEE Properties

It was found that compound **3l** exhibits a pronounced aggregation-induced enhanced emission (AIEE) behavior (Figure 8a). With an increase of the water fraction (f_w_) in a solution of **3l** in acetonitrile, a new emission band appeared with a maximum of 468 nm. The highest intensity was achieved at f_w_ = 60%, while the further increase in f_w_ slightly decreased the emission intensity. This AIEE behavior is due to the presence of a triphenylamine moiety that realizes the restricted intramolecular rotation (RIR) mechanism, whereas fluorophores with any other groups at both *para*- and *meta*-positions did not demonstrate any AIEE activity, except **3b**. The **3b** molecule exhibited unusual behavior in the water-acetonitrile mixture. In this case, the emission maximum shifted from 482 nm to 492 nm at f_w_ = 10%, accompanied by the quenching of fluorescence. However, a bathochromic shift from 460 nm to 492 nm was detected starting from f_w_ = 70%, accompanied by an enhancement of the fluorescent emission, which reached a limit at f_w_ = 80% (Figure 8b). This peculiar behavior at f_w_ = 0–70% could be explained by the high polarity of water and at f_w_ = 70–90%, by the formation of aggregates.

### 2.6. Studies of Sensing Properties

Electron-rich π-conjugated azine fluorophores can act as sensors or probes for high electron-deficient nitroaromatic explosives via exhibiting photoinduced electron transfer (PET) from one molecule to another [39]. This process is accompanied by the quenching of the fluorescence of a sensor.

In this regard, fluorophores **3a**,**h** were chosen among the whole series of α-(*N*-biphenyl)-substituted 2,2′-bipyridine as model objects in the study of the sensory properties of these fluorophores in the detection of nitroaromatic explosives. The position of the phenyl ring in the aniline residue is the only difference between **3a** (*meta*) and **3h** (*para*) that will allow their comparative analysis. Picric acid (PA) and 2,4,6-trinitrotoluene (TNT) were chosen as the most typical analytes. To determine the efficiency of fluorescence quenching, the Stern–Volmer model was chosen as a well-known method for evaluating sensory properties. Fluorometric titration was performed by the single point methodology using a Horiba FluoroMax 4.

Based on the results of the fluorometric titration, Stern–Volmer plots (Figure 9) with corresponding quenching constants K_SV_ (Table 4, Equation (3)), as well as limits of detection (LOD) of nitroanalytes (Table 4, Equation (4)) were determined. Samples **3a**,**h** showed high K_SV_ (from 10.6 to 68.3 M^−1^) and LOD (64.9 to 5535.5 ppb) values, which are in the same order as the described pyridine-based probes [40]. Probe **3a** demonstrated higher K_SV_ values and better sensitivity toward both PA and TNT resulting in a *meta*-position that is more promising for creating sensors and probes for nitroexplosive detection. Thus, α-(*N*-biphenyl)-substituted 2,2′-bipyridines may find a potential application as sensors/probes for the determination of nitroaromatic explosives.
(3)$$\frac{I_{0}}{I}=1+K_{SV}\times \left[Q\right]$$
where *I*_0_ is the fluorescence intensity in the absence of a nitroexplosive, *I* is the fluorescence intensity in the presence of a nitroexplosive, *K_sv_* is the Stern–Volmer constant, [*Q*] is the concentration of a nitroexplosive.

Equation (3). Stern–Volmer equation.
(4)$$LOD=\frac{3\times \sigma }{k}$$
where *LOD* is the limit of detection, *σ* is the standard deviation of the fluorophore intensity in the absence of a nitroexplosive, *k* is the slope of the linear calibration curve.

Equation (4). LOD calculation.

## 3. Materials and Method

### 3.1. Materials and Equipment

Unless otherwise indicated, all common reagents and solvents were used from commercial suppliers without further purification. 6-Phenyl-3-(pyridin-2-yl)-1,2,4-triazine-5-carbonitrile **2a** and 6-(4-tolyl)-3-(pyridin-2-yl)-1,2,4-triazine-5-carbonitrile **2b** were synthesized according to reported literature [26,41].

Melting points were determined on Boetius combined heating stages. TLC and column chromatography were carried out on SiO_2_. ^1^H NMR and ^13^C NMR spectra were recorded at room temperature at 400 and 100 MHz, respectively, on a Bruker DRX-400 spectrometer using CDCl_3_ or DMSO-*d_6_* as the solvent. Hydrogen chemical shifts were referenced to the hydrogen resonance of the corresponding solvent (DMSO-*d_6_*, δ = 2.50 ppm or CDCl_3_, δ = 7.26 ppm). Carbon chemical shifts were referenced to the carbon resonances of the solvent (CDCl3, δ = 77.16 ppm). Peaks were labeled as singlet (s), doublet (d), triplet (t), doublet of doublets (dd), doublet of doublets of doublets (ddd), and multiplet (m). Mass spectra were recorded on a MicrOTOF-Q II (Bruker Daltonics), electrospray as a method of ionization UV–vis absorption spectra were recorded on a Shimadzu UV-1800 spectrophotometer, and emission spectra were measured on a Horiba FluoroMax-4 by using quartz cells with a 1 cm path length at room temperature. Absolute quantum yields of the luminescence of target compounds in solution were measured by using the integrating sphere Quanta-φ of the Horiba FluoroMax 4 at room temperature.

### 3.2. General Method for the Synthesis of 5-Arylamino-1,2,4-Triazines ***1a**–**c*** and ***5a**–**c***

The mixture of corresponding 5-cyano-1,2,4-triazine **2a**,**b** (1 mmol) and corresponding aniline (1.05 mmol) was stirred at 200 °C for 8 h under an argon atmosphere. The products were used in the next step without additional purification. Analytical samples were obtained by flash chromatography with chloroform as eluent.

### 3.3. General Method for the Synthesis of N-Aryl-1-(Pyridine-2-yl)-6,7-Dihydro-5H-Cyclopenta[c]pyridin-3-Amines ***3a**–**l*** and ***6a**–**c***

The mixture of corresponding 5-arylamino-1,2,4-triazine **1a**–**c** (0.3 mmol) or **5a**–**c** and 1-morpholinocyclopentene (1.5 mmol) was stirred at 200°C for 2 h under an argon atmosphere. Then, the additional portion of 1-morpholinocyclopentene (0.75 mmol) was added and the resulting mixture was stirred for an additional 1 h at the same conditions. The products were separated by flash chromatography (DCM as eluent) and were then purified by recrystallization (ethanol).

### 3.4. General Method for the Synthesis N-Aryl-1-(Pyridine-2-yl)-6,7-Dihydro-5H-Cyclopenta[c]pyridin-3-Amines ***3a**–**l*** Via Suzuki Cross-Coupling Reaction

The corresponding compound **6a**–**c** (0.3 mmol) was dissolved in THF (15 mL), followed by the addition of the corresponding boronic acid (0.33 mmol), Pd(tpp)_2_Cl_2_ (6.3 mg, 0.009 mmol), triphenylphosphine (3.9 mg, 0.015 mmol), and solution of K_2_CO_3_ (415 mg, 3 mmol) in water (15 mL). The resulting mixture was stirred at 90 °C in an argon atmosphere for 24 h. After completion, the reaction mixture was extracted with ethyl acetate (10 mL). The organic phase was washed with an aqueous solution of KOH, ammonium chloride and water, and dried with anhydrous sodium sulfate. The solvent was removed under reduced pressure. Ethanol was added to the residue and the obtained precipitate was filtered off, washed with ethanol, and dried.

### 3.5. Crystallography

The XRD analysis was carried out using equipment at the Center for Joint Use “Spectroscopy and Analysis of Organic Compounds” at the Postovsky Institute of Organic Synthesis of the Russian Academy of Sciences (Ural Branch). The experiments were accomplished on an automated X-ray diffractometer Xcalibur 3 with CCD detector using a standard procedure (MoKα-irradiation, graphite monochromator, ω-scans with 1o step at T = 295(2) K). Empirical absorption correction was applied. The solution and refinement of the structures were accomplished using the Olex program package [42]. The structures were solved by the method of the intrinsic phases in ShelXT program and refined by ShelXL by the full-matrix least-squared method for non-hydrogen atoms [43]. The H-atoms at C-H bonds were placed in the calculated positions, and the H-atoms at N-H bonds were refined independently in isotropic approximation.

The result of the X-ray diffraction analysis for compound **3a** was deposited with the Cambridge Crystallographic Data Centre (CCDC 2167697) and can be found in supporting documents.

## 4. Conclusions

In conclusion, a rational synthetic approach toward α-biphenylamino-2,2′-bipyridines push–pull fluorophores starting from 5-cyano-1,2,4-triazines precursors using the sequence of the ipso-nucleophilic aromatic substitution of the cyano group, aza-Diels–Alder reaction and Suzuki cross-coupling was reported. In the THF solutions, the herein-reported fluorophores exhibited an intensive emission with a maxima of up to 511 nm along with up to 49.1% quantum yields. The most representative fluorophores **3b**,**g**,**i**,**l** exhibited a well-pronounced positive fluorosolvatochromic behavior, which was well correlated with the solvent polarity scales according to Kosower and Dimroth/Reichardt as well as Lippert–Mataga mathematical models. DFT calculations confirmed the presence of the ICT state of the fluorophores. Fluorophores **3l** and **3b** demonstrated an unexpected AIEE activity, while fluorophores **3a** and **3h** exhibited sensing properties toward nitroaromatics (explosives) with high Ksv and LOD values.

## Data Availability

The data presented in this study are available on request from the corresponding author and co-authors.

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
