# Peer review of "Synthesis and Photophysical Properties of α-(N-Biphenyl)-Substituted 2,2′-Bipyridine-Based Push–Pull Fluorophores"

_molecules, 2022, doi:10.3390/molecules27206879_

Round 1

Reviewer 1 Report

In the manuscript entitled "Synthesis and photophysical properties of α-(N-biphenyl)-substituted 2,2'-bipyridine-based push-pull/AIE-fluorophores" The authors continue their work on the synthesis and photophysical studies of α-biphenylamino-2,2'-bipyridines. They tested a broader scope of substrates using a different approach with good yields. Although the photophysical studies are comprehensive, the compounds are characterized by NMR and the purity observed across the NMR data provided is good. However, I would NOT recommend publishing this manuscript in Molecules in its current form.

I would recommend that a revised manuscript may become acceptable for publication in Molecules after addressing some of the questions and comments outlined below:

Comments/Questions:

1. The reviewer cannot access the X-ray data of 3a via CCDC based on the CCDC 2167697 the author provided.

2. In line 238, the authors mentioned the energy gaps correlate well with the UV data, the optical gaps for these compounds estimated from the UV spectra should be provided.

3. Please indicate the solvent model (SMD? PCM?) which is used in DFT calculations. Also, there are no computational details for excitation state calculations.

4. To show the charge transfer process, the electron density difference maps (EDD) of the ground state and the excited state of interest would be more beneficial instead of molecular electrostatic potential (MEP) distribution herein.

5. The caption in Figure 8b, ‘a plot of fluorescence intensity of 3i at 468 nm at different fw value.’ The plot intensity should be 482 nm for 3i

6.  In the AIE behavior section, the author stated that ‘this peculiar behavior (for 3i) could be explained by the meta-substitution of the triphenylamine moiety and less effective RIR’, In this case of 3i, the emission maximum shifted from 482 nm to 492 nm 258 at fw = (10% to 70%) accompanied by quenching of fluorescence, which is probably due to the increasing of solvent polarity and the non-radiative decay rate increased, while the latter part starting from fw = 70% to 80% could be AIE effect. Please note the intermolecular interactions (such as H-bond, as the author also indicates in the Lippert-Mataga plot section) with water could also produce usual behavior.

Does compound 3l emit light in the solid state?

7. The raw data for fluorimetric titration should be provided.

*ESI section:

1. Could the author assign the unintegrated singlet peaks around 8.10 to 8.18 ppm in each 1H-NMR spectrum for compounds 5, 6

Author Response

  1. The reviewer cannot access the X-ray data of 3a via CCDC based on the CCDC 2167697 the author provided.

Answer: Thank for your comment. We have checked CCDC website. The X-ray data for 3a are available now

  1. In line 238, the authors mentioned the energy gaps correlate well with the UV data, the optical gaps for these compounds estimated from the UV spectra should be provided.

Answer: We are grateful to the Reviewer 1 for his suggestion. The energy gaps calculated from the UV/Vis data were added into Supplementary file.

  1. Please indicate the solvent model (SMD? PCM?) which is used in DFT calculations. Also, there are no computational details for excitation state calculations.

Answer: Thank you for your comment. The solvent model was SMD, and we have indicated this fact in the revised version of our manuscript. In addition, we have added Computational details section concerning all our calculations in Supplementary Materials.

  1. To show the charge transfer process, the electron density difference maps (EDD) of the ground state and the excited state of interest would be more beneficial instead of molecular electrostatic potential (MEP) distribution herein.

Answer: We are grateful to the Reviewer for this valuable comment. The electron density difference maps (EDD) were added in Supplementary Materials.

  1. The caption in Figure 8b, ‘a plot of fluorescence intensity of 3i at 468 nm at different fw value.’ The plot intensity should be 482 nm for 3i.

Answer: The Figure 8b was updated according to the comment.

  1. In the AIE behavior section, the author stated that ‘this peculiar behavior (for 3i) could be explained by the meta-substitution of the triphenylamine moiety and less effective RIR’, In this case of 3i, the emission maximum shifted from 482 nm to 492 nm 258 at fw = (10% to 70%) accompanied by quenching of fluorescence, which is probably due to the increasing of solvent polarity and the non-radiative decay rate increased, while the latter part starting from fw = 70% to 80% could be AIE effect. Please note the intermolecular interactions (such as H-bond, as the author also indicates in the Lippert-Mataga plot section) with water could also produce usual behavior.

Answer: We thank the reviewer for raising this point. First of all, the correct number of the compound is 3b, not 3i (which was a mistake). Indeed, water can increase the solvent polarity and red-shift fluorescence emission. We have updated this section according to this explanation. However, water did not influence on fluorescence of similar fluorophore 3l (except formation of aggregates), and this behavior of 3b is still discussible.

Does compound 3l emit light in the solid state?

Answer: Unfortunately for us, among 3a-l compounds no one can emit light in solid state, including 3l.

  1. The raw data for fluorimetric titration should be provided.

Answer: We greatly appreciate the Reviewer for the comment The fluorometric titration was performed by the single-point methodology using Horiba FluoroMax 4. We have added this information to the Manuscript.

*ESI section:

  1. Could the author assign the unintegrated singlet peaks around 8.10 to 8.18 ppm in each 1H-NMR spectrum for compounds 5, 6?

Answer: We are thankful the Reviewer 1 for mentioning this fact. This is a solvent peak of the proton of chloroform in DMSO-d6, which does not belong to a structure of the synthesized compounds.

Reviewer 2 Report

In this contributuon, Zyryanov and coworkers designed a series of twelve a-(N-biphenyl)substituted 2,2’-bipyridine derivatives. I addition to the synthesis of target compounds, photophysical properties are described including for selected compounds emission solvatochromism, effect of aggregation on emission, emission quenching in the presence of nitroaromatic exoplosives. Basic DFT calculation are also described for selected compounds. This work presents some interest and fits the scope and level of Molecules journal. Nevertheless some major issues need to be adressed before acceptance :

1-     N-biphenyl : N  should be in itallic : please correct in the title and in many place in the manuscript.

2-     Compound 3l exhibit AIEE (agregation induced enhanced emission) not AIE : significante mission is already observed in pure MeCN. Only for one compound AIEE is demonstrated : I would suggest to remove it from the title on the manuscript.

3-     I am not sure that « 1,2,4-triazine methodology » is an adapted keyword

4-     Figure 1 : please associate each structure with a littérature reference.

5-     When talking about ipso-substitution, it should be mentioned that it is in fact an ipso-nucleophilic aromatic substitution.

6-     L 58 : what means TM-free ? all abbrevation should be defined at 1st use.

7-     Scheme 1 : in the caption Synthetic routes. R1, R2, R3 instead of R1, R2, R3 . For compound 2a,b :  it should be R1 (or better R to avoid to have two R1) instead of the methyl group. In the transformation of 6 to 3 : PPh3 instead of tpp

8-     L 95-96 : It is written : « The bond distances and angles in the molecules are near standard values ». At least a tables with selected bond distances and angles should be provided in ESI.

9-     L 123/l126 : 1,4-phenylene fragments instead of benzene fragment/aromatic substituents.

10-  Table 1 : The solvent should be specified in the caption. Stokes shifts should be expressed as wavenumber difference (in cm-1). Even if measured with an integration sphere, the PLQY should be provided with only 2 significant digits (eg 19% instead of 19.4%). Molar absorption coefficient should be provided in addition to absorption maxima.

11-  Regarding the calculation of Stokes shifts, considered absorption and emission band should correspond to the same excited state. If the emission correspond to HOMO-LUMO transition, in this case if a shoulder at higher energy is observed, this is this value that should be concidered. For instante the Stokes shifts of 3e  is of 100 nm not of 450 nm. Please correct it carrefully.

12-  Figure 3 part a : It would be better to plot the molar absorption coefficient versus wavelength. Spectra could begin at 250 nm (the 200-250 part is not significant). The concentration (or concentration range) should be indicated din the caption.

13-  When talking about well-pronnonced solvatochromic behavior, the authors refer to emission solvatochromism (or fluorosolvatochromism. Please correct in various place of the manuscript.

14-  P 168 it is written « a is the radius of the cavity in which the fluorophore resides ». What is the value choosen for a and how was it calculated estimated ? a littérature reference is required here. A slight modification of a would dramadically modify the Dµ value

15-  Compound number should be systematiclaly in bold withing the manucript, including in tables, figures and their caption.

16-  The inversion of solvatochromism in the most polar solvent is strange. I think the case of MeOH can be easilly explained by the protic character of this solvent but for othe rI am wondering if it is not due to the presence of water in DMF and DMSO used that would artificaly increase the polarity.

17-  L 227 : It is not clear how the solvent effect was taken into account : PCM model ?

18-  It is not clear how the differences of dipole moment between ground and excited states were  calculated. The geometry of the compounds in excited state was calculated ? It so, it should be discussed.

19-  L 261-262 : It is written that « This peculiar behavior could be explained by the meta-substitution of the triphenylamine moiety and less effective RIR » Is it 3i of 3b that was studied ? this is confusing. It fact this is the behavior obserbed for 3i (or 3b?) tha tis expected : 1st a red shift of emission due to increase of polarity of solvent mixcture due to increase of water ratio then a blue shift is observed due to the formation of agregate in whcih the polarity is reducted. This is more the red-shift observed upon aggregation for 3l that is surprising and need to be explained.

20-  Table 1 entry 18 C6H4 (digits in subscript)

21-  Figure 9 should be removed : the luminescence of the cuvette (not in quartz) and the top makes the picture not useful.

22-   In ESI please replace the number of mol obtained by the mass obtained. 13C nmr chemical shifts should be provided with only one decimal.

Author Response

  • N-biphenyl: N should be in italic: please correct in the title and in many place in the manuscript.

Answer: We are thankful the Reviewer 2 for mentioning this fact. The Manuscript was updated according this comment.

  • Compound 3l exhibit AIEE (agregation induced enhanced emission) not AIE : significante mission is already observed in pure MeCN. Only for one compound AIEE is demonstrated : I would suggest to remove it from the title on the manuscript.

Answer: We agree with the Reviewer and have updated the Manuscript and its title.

  • I am not sure that « 1,2,4-triazine methodology » is an adapted keyword

Answer: We have removed this keyword and believe that the current list of keywords can fully characterize this research work.

4-     Figure 1 : please associate each structure with a littérature reference.

Answer: Required references were added into Figure 1.

5-     When talking about ipso-substitution, it should be mentioned that it is in fact an ipso-nucleophilic aromatic substitution.

Answer: We agree with the Reviewer. The Manuscript was updated according the comment.

 6-     L 58 : what means TM-free ? all abbrevation should be defined at 1st use.

Answer: TM-free means “transition metal free”. The Manuscript was updated according the comment.

7-     Scheme 1 : in the caption Synthetic routes. R1, R2, R3 instead of R1, R2, R3 . For compound 2a,b :  it should be R1 (or better R to avoid to have two R1) instead of the methyl group. In the transformation of 6 to 3 : PPh3 instead of tpp

Answer: The Scheme 1 was updated

8-     L 95-96 : It is written : « The bond distances and angles in the molecules are near standard values ». At least a tables with selected bond distances and angles should be provided in ESI.

Answer:  Selected bond lengths and angles for compound 3a were added into supporting file.

9-     L 123/l126 : 1,4-phenylene fragments instead of benzene fragment/aromatic substituents.

Answer: We thank the Reviewer for the comment. The Manuscript was updated.

10-  Table 1 : The solvent should be specified in the caption. Stokes shifts should be expressed as wavenumber difference (in cm-1). Even if measured with an integration sphere, the PLQY should be provided with only 2 significant digits (eg 19% instead of 19.4%). Molar absorption coefficient should be provided in addition to absorption maxima.

Answer: Table 1 was updated according the comment.

11-  Regarding the calculation of Stokes shifts, considered absorption and emission band should correspond to the same excited state. If the emission correspond to HOMO-LUMO transition, in this case if a shoulder at higher energy is observed, this is this value that should be concidered. For instante the Stokes shifts of 3e  is of 100 nm not of 450 nm. Please correct it carrefully.

Answer: We grateful the Reviewer for the provided comment. Table 1 and the text in the section 2.2 “Photophysical studies” was updated.

12-  Figure 3 part a : It would be better to plot the molar absorption coefficient versus wavelength. Spectra could begin at 250 nm (the 200-250 part is not significant). The concentration (or concentration range) should be indicated din the caption.

Answer: We grateful the Reviewer for the provided comment. Figure 3 and Table 1 were updated according to the comment

13-  When talking about well-pronnonced solvatochromic behavior, the authors refer to emission solvatochromism (or fluorosolvatochromism. Please correct in various place of the manuscript.

Answer: We are grateful to the Reviewer 2 for this clarification. Indeed, fluorosolvatochromism is more correct term and we updated the revised version of our manuscript.

14-  P 168 it is written « a is the radius of the cavity in which the fluorophore resides ». What is the value chosen for a and how was it calculated estimated ? a littérature reference is required here. A slight modification of a would dramadically modify the Dµ value

Answer: We selected a as 4Å according to the [Lakowicz, J.R. Principles of Fluorescence Spectroscopy; Lakowicz, J.R., Ed.; 3rd ed.; Springer US: Boston, MA, 2006] as the most representative value and have clarified this in our revised manuscript

15-  Compound number should be systematically in bold withing the manuscript, including in tables, figures and their caption.

Answer: This was corrected in our revised manuscript

16-  The inversion of solvatochromism in the most polar solvent is strange. I think the case of MeOH can be easily explained by the protic character of this solvent but for other I am wondering if it is not due to the presence of water in DMF and DMSO used that would artificially increase the polarity.

Answer: We thank the Reviewer for raising this point. Such deviations can not be explained by traces of water in selected high polar solvents due to the absence of these deviations in related α-arylamino-2,2'-bipyridines [Kopchuk, D.S.; Krinochkin, A.P.; Starnovskaya, E.S.; Shtaitz, Y.K.; Khasanov, A.F.; Taniya, O.S.; Santra, S.; Zyryanov, G. v.; Majee, A.; Rusinov, V.L.; et al. 6-Arylamino-2,2′-Bipyridine “Push-Pull” Fluorophores: Solvent-Free Synthesis and Photophysical Studies. ChemistrySelect 2018, 3, 4141–4146, doi:10.1002/slct.201800220] (the ones without the 1,3- or 1,4-phenylene unit). However, the protic nature of methanol could influence on corresponding emission spectra. We have updated the revised version of our manuscript.

17-  L 227 : It is not clear how the solvent effect was taken into account : PCM model ?

Answer: No, the solvent model was SMD, and we added this information with appropriate reference in the Computational details section in Supplementary Materials.

18-  It is not clear how the differences of dipole moment between ground and excited states were  calculated. The geometry of the compounds in excited state was calculated ? It so, it should be discussed.

Answer: Yes, and appropriate clarification was added in the revised version of
our manuscript.

19-  L 261-262 : It is written that « This peculiar behavior could be explained by the meta-substitution of the triphenylamine moiety and less effective RIR » Is it 3i of 3b that was studied ? this is confusing. It fact this is the behavior observed for 3i (or 3b?) that is expected : 1st a red shift of emission due to increase of polarity of solvent mixture due to increase of water ratio then a blue shift is observed due to the formation of aggregate in which the polarity is reduced. This is more the red-shift observed upon aggregation for 3l that is surprising and need to be explained.

Answer: We thank the Reviewer 2 for raising this point. Indeed, the correct number of the compound is 3b (not 3i) and we are grateful to the Reviewer for discovering this misprint. Water can increase the solvent polarity and red-shift fluorescence emission. We have updated this section in the revised version of our manuscript. However, water did not influence on fluorescence of similar fluorophore 3l (except formation of aggregates), and this behavior of 3b is still discussible.

20-  Table 1 entry 18 C6H4 (digits in subscript)

Answer: We have corrected this.

21-  Figure 9 should be removed : the luminescence of the cuvette (not in quartz) and the top makes the picture not useful.

Answer: We are thankful to the Reviewer 2 for this suggestion. We have updated the revised version of our manuscript according to this.

22-   In ESI please replace the number of mol obtained by the mass obtained. 13C nmr chemical shifts should be provided with only one decimal.

Answer: The ESI file was updated.

Round 2

Reviewer 1 Report

The authors have responded to the comments point by point very clearly.

However, the EDD maps in FigureS9 might not be correct, I was surprised to say there is a minimal charge transfer in such an ICT fluorophore like 3l. It seems to me that these EDD maps only contain the Total SCF Density for both states. One example for EDD map would be: 10.1039/C7SC01997A

To calculate the EDD map,

For Gaussain09, one should do a TD-DFT calculation based on the S0 structure and write the density function into the chk file by using' density' keywords.

For more details, please refer to:

James B. Foresman and AEleen Frisch.Plotting an Electron Difference Density. Exploring Chemistry with Electronic Structure Methods. Third Edition. Page:338

In fact,  the HOMO-LUMO plots in Figure 7 have already shown the ICT effects as the author indicated. They may delete FigureS9 if they would have some issues with the EDD maps.

Author Response

We thank the Reviewer for the thorough critique of the manuscript. Due to the contradictory of the EDD data, Figure S9 and the corresponding explanation have been removed

Reviewer 2 Report

The manuscript has been significantly improved. 

In  Table 1: Stokes shifts should be provided with reasonable number of significant digits: these data are calculated starting from wavelength in cm-1: please provide the value with only 3 significant digits (eg 5830 cm-1 instead of 5827.06 cm-1).

Author Response

We thank the Reviewer for this remark. Table 1 was updated.
